# Integrated taxonomy of black flies (Diptera: Simuliidae) reveals unexpected diversity in the most arid ecosystem of Europe

Ignacio Ruiz-Arrondo[1]*, Jesús Veiga[2,3,4], Peter H. Adler[5], Francisco Collantes[6], José A. Oteo[1], Francisco Valera[2]

1 Center of Rickettsiosis and Arthropod-Borne Diseases (CRETAV), Infectious Diseases Department, San Pedro University Hospital-Center for Biomedical Research from La Rioja (CIBIR), Logroño, La Rioja, Spain, 2 Departamento de Ecología Funcional y Evolutiva, Estación Experimental de Zonas Áridas (EEZA-CSIC), Almería, Spain, 3 Departamento de Parasitología, Facultad de Farmacia, Universidad de Granada, Granada, Spain, 4 MEMEG, Department of Biology, Lund University, Lund, Sweden, 5 Department of Plant and Environmental Sciences, Clemson University, Clemson, South Carolina, United States of America, 6 Departamento de Zoología y Antropología Física, Facultad de Biología, Universidad de Murcia, Murcia, Spain

* irarrondo@riojasalud.es

**Data Availability Statement:** All relevant data are within the paper and its Supporting Information files.

## Abstract

The family Simuliidae includes more than 2000 species of black flies worldwide. Their morphological uniformity creates difficulty for species identification, which limits our knowledge of their ecology and vectorial role. We investigated the systematics of black flies in a semi-arid area of the Iberian Peninsula, an ecologically harsh environment for these organisms. Sampling adult black flies in three different habitats (by means of CDC traps) and in avian nest boxes and collecting immature stages in high-salinity rills provided a representative sample of the component species. A combination of approaches, including morphological, chromosomal, and molecular (based on the mitochondrial cytochrome C oxidase subunit I (COI) and internal transcribed spacer 2 (ITS2) genes) revealed five species: four common species (*Simulium intermedium*, *S. petricolum*, *S. pseudequinum*, and *S. rubzovianum*) and the first European record for *S. mellah*. Barcoding gap and phylogenetic analyses revealed that ITS2 is a key marker to identify the species, whereas the COI marker does not provide enough resolution to identify some species or infer their phylogenetic relationships. Morphological and chromosomal features are also provided to identify *S. mellah* unequivocally. Our study highlights the need for integrated studies of black flies in ecologically extreme habitats to increase our knowledge of their distribution, ecology, and potential risks for public health.

## 1. Introduction

The Iberian Peninsula, between North Africa and southern Europe, is part of one of the world's biodiversity hotspots [1]. European, Iberian endemic, and Ibero-African/Maghrebian taxa occur here, reflecting late Miocene (Messinian) events [2, 3]. The Iberian Peninsula is also

**Funding:** This study was financially supported by National Institute of Food and Agriculture (NIFA) and United States Department of Agriculture (USDA), under project number SC-1700596, with Technical Contribution No. 7066 of the Clemson University Experiment Station in the form of an award received by PHA. This study was also financially supported by Ministerio de Ciencia e Innovación (MICIN), Agencia Estatal de Investigación ((AEI)/10.13039/501100011033) and by "ERDF A way of making Europe" in the form of awards (CGL2014-55969 and PGC2018-097426-B-C22) received by FV. This study was also financially supported by a predoctoral grant (BES-2015-075951) from the Spanish Ministry of Science and Innovation received by JV. This study was also financially supported by a postdoctoral grant "Margarita-Salas" funded by the Spanish Ministry of Universities and the European Union – NextGenerationEU received by JV. This study was also financially supported by the postdoctoral grant "Juan de la Cierva-formación" (FJC2021-048057-I) received by JV funded by Ministerio de Ciencia e Innovación (MCIN)/Agencia Estatal de Invetigación (AEI)/10.13039/501100011033 and by European Union – NextGenerationEU/Planes de Recuperación, Transformación y Resiliencia (PRTR)". The funders had no role in study design, data collection and analysis, decision to publish, or preparation of the manuscript.

**Competing interests:** The authors have declared that no competing interests exist.

a hotspot for black flies (Diptera: Simuliidae) [4], representing the northern geographic limit for North African species and the southern limit for European species.

The family Simuliidae currently has 2398 formally described, living species worldwide, and new species are added each year [5]. However, black flies are morphologically difficult to identify [6]. Molecular techniques offer a valuable approach for identification but have limitations in some groups [7–9]. Therefore, an integrated taxonomic approach combining morphological, chromosomal, and molecular techniques is necessary to identify and understand the diversity of species [9, 10], particularly in hotspots such as the Iberian Peninsula, where 49 species of black flies are known [5], several of them recently discovered in chromosomal studies [11].

Black flies are one of the most important groups of bloodsucking vectors, capable of transmitting filarial nematodes, trypanosomes, haemosporidian parasites, and vesicular stomatitis viruses [12, 13]. Vector-borne parasites play a major role in epidemiological, evolutionary, and ecological processes [14, 15]. The distribution of vector-borne parasites depends on factors such as the availability of vectors, abilities of host species to maintain the parasites, and habitat choice and behavior of the vectors [16, 17]. Although we know much about some vectors (e.g., mosquitoes) in some areas [18], our knowledge of other vectors is often limited. The identity of interacting organisms is also frequently ignored, particularly for hosts and their parasites [19]. Consequently, we lack information about key epidemiological processes, the potential of pathogen transmission and disease risk and, thus, the appropriate disease control approaches [20]. Knowledge gaps are particularly evident for black flies. Many studies are based on the preimaginal stages, in part because of the accessibility of breeding sites [21]. Studies of adult simuliids in Europe are limited despite interest in obtaining a complete picture of the avian blood-parasite community [22–24] and of the impact of simuliids on public health [21, 25, 26]. An additional bias in our knowledge, given that black flies are linked to flowing water, is that many studies have been conducted in favorable ecosystems such as wet mountainous areas [27–29]. Yet, black flies also dwell in arid lowlands [6, 30–33]. Different trapping and identification methods and selection of specific habitats for sampling are needed to provide a comprehensive view of the simuliid community in any area.

Spain has numerous semi-arid areas, especially in the southeastern region, and a diversity of watercourses, but few studies have been conducted on simuliids in semi-arid areas [34] and those that have are based only on the preimaginal forms at a few sampling sites. In Almería, the most arid area in Europe, 10 species of black flies have been found [34–38], whereas in the neighboring province of Murcia, only three species are known [25, 39]. Yet, specific studies of particular habitats suggest an uneven distribution of some species and specialisation by some. For instance, investigations of black flies associated with cavity-nesting wild birds in an arid area in Almería revealed two species (*Simulium rubzovianum* (Sherban) and *S. petricolum* (Rivosecchi)) [32, 40]. These apparent preferences and the scarce information on the black fly community in arid environments suggest that focused studies could highlight important distributional and ecological aspects (e.g., habitat selection) and provide relevant information about host-parasite relationships.

We aimed to provide comprehensive information about the black fly community in the most arid area of Europe by i) sampling in the main habitats (ramblas, sandstone cliffs, and wood patches) and in specific microhabitats (nest boxes of a common wild bird species) to obtain adults, ii) sampling in creeks to collect larvae and pupae, and iii) integrating taxonomic methods (morphological, molecular, and chromosomal). Given that black flies require running water, we anticipated a depauperate simuliid community in this semi-arid environment. We also performed a comparative taxonomic analysis of an insufficiently known species, *S. mellah* Giudicelli & Bouzidi, because it featured prominently in our study, and our collections represent the first record of this species for Europe.

## 2. Material and methods

### 2.1. Study area

The study area (ca. 50 km$^2$) lies in Tabernas Desert (Almería, SE Spain, 37˚05′N 2˚21′W). The climate is temperate, semi-arid Mediterranean with a water deficit during the long, hot summer when the absolute maximum monthly temperature is above 40˚C and the monthly average maximum daily temperature is above 30˚C. The average annual temperature is 18˚C, with mild inter-annual oscillations of 3–4˚C and significant intra-annual fluctuations. The mean annual rainfall is ca. 230 mm, with high inter-annual and intra-annual variability [41].

The landscape consists mostly of patches of shrubby vegetation and olive and almond groves interspersed among dry watercourses (i.e., ramblas). Inhabited farms are scarce. Small, isolated patches of *Eucalyptus* trees are throughout the study area.

The main body of water in the study area is Rambla de Tabernas, with a few springs flowing into it. Rambla de Tabernas carries a series of small surface flows (1–5 liters/second) of saline water that derive from highly saline groundwater and are discontinuous in space but continuous in time [42]. Thus, the water sheet available for aquatic insects is patchy. Evaporation is greatest during dry weather, increasing salinity and depositing salts that are washed downstream during the wetter seasons [42]. Surface water has a sodium chloride facies with high and variable salinity and elevated Cl$^-$, SO$_4^2$, and Na$^+$ [42, 43]. Mean temperature of flowing water in Rambla de Tabernas is 19.4˚C (range: 7.9–32.5˚C) [43]. Four rills at the periphery of the study area (one in Cabeza del Águila at 37˚05'09"N 02˚24'12"W and three in El Cautivo at 37˚00'49"N 02˚26'26"W) were sampled for preimaginal black flies. The two groups of rills were 8.7 km apart in two different ramblas, one about 243 m above sea level (asl) and the other about 468 m asl. These two areas were 1.15 km and 4.3 km from the nearest nest boxes (see below). The four small watercourses were in direct sunlight and had warm, clear water with minimal flow that evaporated after approximately 50–70 m at the time of sampling. Sections of the four rills revealed aggregations of larvae and pupae fixed to a rocky and sometimes sandy bottom. Minimal patches of filamentous algae covered with organic matter were in the rills.

The bird community in the study area includes open-nesting species (small passerines, doves, and pigeons) and cavity-nesting species breeding mainly in natural cavities in ramblas and human-made constructions [44]. Among the cavity-nesting species, the European roller (*Coracias garrulus* L.) and the rock pigeon (*Columba livia* Gmelin) are the most abundant. As a result of a long-term nest-box program, most European rollers (hereafter rollers) currently breed in wooden nest boxes installed in *Eucalyptus* trees, sandstone banks, and human constructions [45]. Rollers are trans-Saharan migratory birds arriving at the breeding grounds in the study area at the end of April. The breeding season (1 brood/year) usually lasts until mid-July. The eggs are incubated for ca. 21 days and fledglings leave the nest 22–25 days after hatching.

### 2.2. Black fly collection

**2.2.1. Collecting adult black flies in nest boxes.** Adult black flies were collected with sticky traps (glue-paper fixed under the upper lid of the nest box [32]) in 37 and 36 nest boxes occupied by rollers in 2017 and 2018, respectively. Sticky traps were placed during the nestling stage in two successive periods. The first period was for 4 days 13–14 days after the first egg hatched. The traps then were replaced by new ones for 4 more days, so that the whole trapping time was 8 days. The trapping period spanned 2 June–18 July in 2017 and 6 June–23 July in 2018. Black flies also were caught opportunistically in the nests by hand during routine visits.

We used a subsample of 17 of 155 adult black flies captured in 2017 and 68 of 515 in 2018. The remaining individuals were used for other purposes [32]. All specimens were fixed in 95% ethanol and held at -20°C until identification.

**2.2.2. Collecting adult black flies with CDC light traps.** During three breeding seasons (2016–2018), CDC light traps were placed in three habitats where rollers and other cavity-nesting birds breed: i) sandstone cliffs, ii) bridges over ramblas, and iii) near isolated trees or groups of trees (mainly *Eucalyptus*). At each sampling point, two CDC traps (one with ultraviolet light and one with incandescent light) were placed ca. 50 cm apart and 2–7 m above ground. Each pair of traps was baited with 1 kg of dry ice per night. Trapping sessions were adjusted according to the breeding season of rollers and the moon calendar [46]; windy nights were avoided.

In 2016 and 2017, 20 sampling points were established: eight on cliffs, eight on trees, and four on bridges over ramblas. In 2016, each point was sampled once; a group of 10 pairs of traps was placed 8–10 June and a second group of 10 pairs of traps 7–8 July. In 2017, all 20 pairs were set from 22 June to 1 July. In 2018, only 16 of the 20 sampling points were established: six on cliffs, six near trees, and four on bridges. These points were sampled twice, 11–14 June and 10–15 July. An additional sampling point in a deserted farmhouse was sampled once (14 June 2018). The sampling points were distributed throughout the study area, with the distance between the farthest points being 22.5 km.

Captured insects were transferred to the Estación Experimental de Zonas Áridas and held at -20°C until identification.

**2.2.3. Collecting larvae and pupae from rills.** Larvae and pupae were hand-collected, using forceps, during three visits (21 May and 13 November 2019, and 7 April 2020). They were fixed in 95% ethanol for morphological and molecular purposes, and additional larvae were fixed in two changes of Carnoy's solution (1 part glacial acetic acid: 3 parts 95% ethanol) for chromosomal study. All samples were held at -20°C until processing.

The authority of Junta de Andalucia provided permits to conduct our fieldwork.

## 2.3. Taxonomic assessments of black flies

**2.3.1. Morphological analyses of preimaginal and adult black flies.** For morphological identification, we cleared specimens (85 females from nest boxes and 37 females and 30 males from CDC traps) before slide-mounting them in Hoyer's medium. All specimens were compared with published descriptions [36, 47–49]. We also compared our specimens with those of other Palearctic species in the *S.* (*Eusimulium*) *aureum* group [37, 48, 50].

To provide a more complete description of the larva, pupa, female, and male of *S. mellah*, we conducted a detailed morphological study. Larvae (n = 32) of the *S. aureum* group from three (El Cautivo) of our four stream sites were the same specimens (ultimate and penultimate instars) that were studied chromosomally. Associated pupae (n = 4) from these sites were also studied, as were pharate males (n = 2) from these pupae and females (n = 2) trapped in nest boxes. All specimens of *S. mellah* were dissected in 80% ethanol, and relevant parts (larval head capsules, adult terminalia, and female hind legs and mouthparts) were transferred to 85% lactic acid and heated in a crucible for ca. 1 minute to remove soft tissues. Dissected parts were transferred to a drop of glycerin in a depression slide, further dissected into component parts, and oriented in the desired view (e.g., dorsal) for imaging. Cocoons and pupal gills were placed directly into glycerin. We photographed structures at multiple focal planes with a Jenoptik ProgRes® SpeedXT Core 5 digital camera mounted on an Olympus BX40 light microscope. We used Helicon Focus (version 7.7.5) stacking software to produce composite images.

**2.3.2. Chromosomal analyses of the *S.* (*Eusimulium*) *aureum* group.** Larvae (n = 32) of the *S. aureum* group collected in Carnoy's solution were presorted into those with dark versus

light head capsules, the former corresponding with the original description of *S. mellah* [47]. Larvae were prepared using the Feulgen procedure, and the stained chromosomes and one gonad (for sex determination) were slide-mounted [51]. Banding patterns of all 32 analyzed larvae were compared with those of photomaps for the *S. aureum* group presented by Adler et al. [6]. Selected chromosomes that included areas of novel rearrangements were photographed, as above, under oil immersion, and the images were imported into Adobe® Photo-Shop® Elements 8 for labelling.

Material used in morphological, chromosomal, and molecular analyses is deposited in the Center for Biomedical Research of La Rioja (CIBIR), Spain, and the Clemson University Arthropod Collection, Clemson, South Carolina, USA.

**2.3.3. Phylogenetic and barcoding gap analyses to identify preimaginal and adult black flies.** A modified Hotshot technique was used for DNA extraction of each individual previously identified morphologically (172 individuals: 8 larvae, 12 pupae, and 152 adults from nest boxes and CDC traps) [9]. For larvae, the posterior of the abdomen was removed and used for DNA extraction. Individuals were not homogenized in the process, so voucher material was recovered after DNA extraction. If PCR amplification failed or if low quantity and quality of DNA was measured by NanoDrop™ (Thermo Fisher Scientific, Waltham, MA, USA), DNA was re-extracted using a DNeasy Blood and Tissue kit (Qiagen, Hilden, Germany).

Molecular analysis included amplification by polymerase chain reaction (PCR) of the mitochondrial cytochrome C oxidase subunit I (COI) and the internal transcribed spacer 2 (ITS2) [52]. COI and ITS2 PCRs were run for the 172 individuals. PCR products were sequenced in both directions using the BigDye R Terminator v3.1 Cycle Sequencing Kit (Applied Biosystems, Forest City, CA, USA) at the Sequencing Unit, CIBIR, Spain. Nucleotide sequences were compared with those in GenBank, using a BLAST algorithm (www.ncbi.nlm.nih.gov/genbank). Additional sequences from the species identified and from related species were obtained from GenBank (S1 and S2 Tables). Eight pupae of *S. rubzovianum* collected previously from other regions of Spain [9] were included in the ITS2 analysis, as there is a limited number of sequences in GenBank. Amplifications and sequencing worked for 143 and 165 individuals in COI and ITS2 analyses, respectively. Sequences were edited on BioEdit 7.2., excluding short sequences and trimming the remaining ones to 632 bp for COI and to 324 bp for ITS2. Thus, 117 COI sequences and 134 ITS2 sequences were used in the phylogenetic analysis. We added 33 COI sequences and six ITS2 sequences of the same or related taxa including one outgroup per gene (*Prosimulium tomosvaryi* (Enderlein) for COI and *Cnephia dacotensis* (Dyar & Shannon) for ITS2) (Supplementary material 1).

Each group of sequences was aligned by the ClustalW algorithm, and a phylogeny was inferred by maximum likelihood (ML). A best-fit nucleotide substitution model was explored for COI and ITS2 alignments and selected based on the Akaike Information Criterion [53]. The best-fit nucleotide substitution model was the "Transversion-Inversion Model 2" (TIM2 +G+I) for the COI alignment and the "3-parameter model" (TPM3u+G) for the ITS2 alignment [54, 55]. These models were used for the phylogeny. To estimate the topology support, bootstrapping based on 1000 replicates was used [56]. Phylogenetic analyses were run with R software 4.1.0 [57], using *Biostrings* [58], *msa* [59], *ape* [60], *phangorn 2.7.1* [61], and *ggplot2* [62].

Species identification efficacy of COI and ITS2 was assessed by barcoding gap analysis, which compares differences between the higher intraspecific distances and the smallest interspecific distance to set a cut-off value for species delimitation [63]. Barcoding gap analyses were performed on the same COI and ITS2 sequences used for phylogenetic analysis, excluding unidentified individuals, outgroups, and singletons. Thus, 140 and 124 sequences from COI and ITS2 genes, respectively, were used. We computed a matrix of pairwise distances

using the Kimura 2-parameter (K2P) models with the *sppDistMatrix* function from the R package *SPIDER v1.3–0* [64]. We then performed a barcoding gap analysis and threshold calculations with the "localMinima" function, which determines possible thresholds, creating a density object from the distance matrix to infer where a dip in the density of genetic distances indicates the transition between intra- and interspecific distances without previous knowledge of species identity [64]. We also evaluated and adjusted the threshold calculations based on the "threshOpt" function that, using the species identity, explores how different threshold values affected the cumulative error of identifications. Specifically, this function returns the number of true positive, false negative, false positive, and true negative identifications at a given threshold, plus the cumulative error (false negative + false positive), setting the optimum threshold where the cumulative errors were minimal. As barcoding gap analysis is strongly affected by threshold values, we also performed the analysis using *TaxonDNA*, which calculates the threshold as a value of intraspecific distances below which 95% of all intraspecific distances are found [65]. Identification accuracy was calculated using the "best close match" (BCM) method of Meier et al. [65]. This function returns an identification status of each individual, based on the calculated threshold as follows: i) "correct", when the name of the closest match is the same; ii) "incorrect", when the name of the closest match is different; iii) "ambiguous", when more than one species is the closest match or is within the given threshold; and iv) "no id", when no species are within the threshold distance. Additionally, nucleotide and haplotype diversity was calculated based on Nei [66] and Nei & Tajima [67] respectively.

## 3. Results

### 3.1. Morphology of the simuliid community

Individuals captured with CDC traps (n = 67) belonged to five species: *Simulium intermedium* Roubaud (2 females), *S. mellah* (1 female), *S. petricolum* (1 female), *S. pseudequinum* Séguy (2 females), and *S. rubzovianum/S. velutinum* (Santos Abreu) (31 females and 30 males). In contrast, only two morphospecies of the *S. aureum* group, identified based on terminalia, were found in the nest boxes (n = 85 females): i) *Simulium rubzovianum /velutinum* (14 in 2017 and 66 in 2018); and ii) *S. mellah* (3 in 2017 and 2 in 2018). A detailed study of female structures of potential taxonomic value for this second morphospecies (Fig 1) revealed that pigmentation extended onto the base of the spermathecal duct (Fig 1A and 1D), as in *S. petricolum* [48]. However, other characters, such as the shape of the hypogynial valves (Fig 1E), hind basitarsus, and claw (Fig 1F), did not match *S. petricolum* as described by Rivosecchi et al. [49] but fit the description of *S. mellah*.

The specimens identified as *S. mellah* (n = 5), based on the shape of the hypogynial valves (Fig 1E) and the pigmented base of the spermathecal duct (Fig 1D), agree with figures of this species given by Belqat & Dakki [48]. The base of the spermathecal duct is pigmented in *S. mellah* and *S. petricolum*, but unpigmented in *S. rubzovianum* and *S. velutinum*. The hypogynial valves of *S. mellah* are subparallel along their inner margins (Fig 1E), forming a narrower space than the more divergent hypogynial valves of *S. petricolum*, *S. rubzovianum*, and *S. velutinum*, which form a 'U' or 'V' shape. The genital fork, anal lobe, and cercus of *S. mellah* (Fig 1A–1C) are similar to those of other members of the *S. aureum* group. The claw of *S. mellah* (Fig 1F) resembles that of *S. velutinum* and *S. petricolum*, as presented by Rivosecchi [50] and Belqat & Dakki [48]. The sensory vesicle in the third maxillary palpomere of *S. mellah* (Fig 1G) resembles that of *S. petricolum* and *S. rubzovianum* (as *S. latinum*) shown by Rivosecchi [50]. The mandible of *S. mellah* (Fig 1H) in our specimens has fewer serrations (19 or 20) than that described by Rivosecchi [50] for *S. rubzovianum* (as *S. latinum*), which was stated to have 22 (although more than 30 serrations are shown in the illustration), and *S. petricolum*, with 23

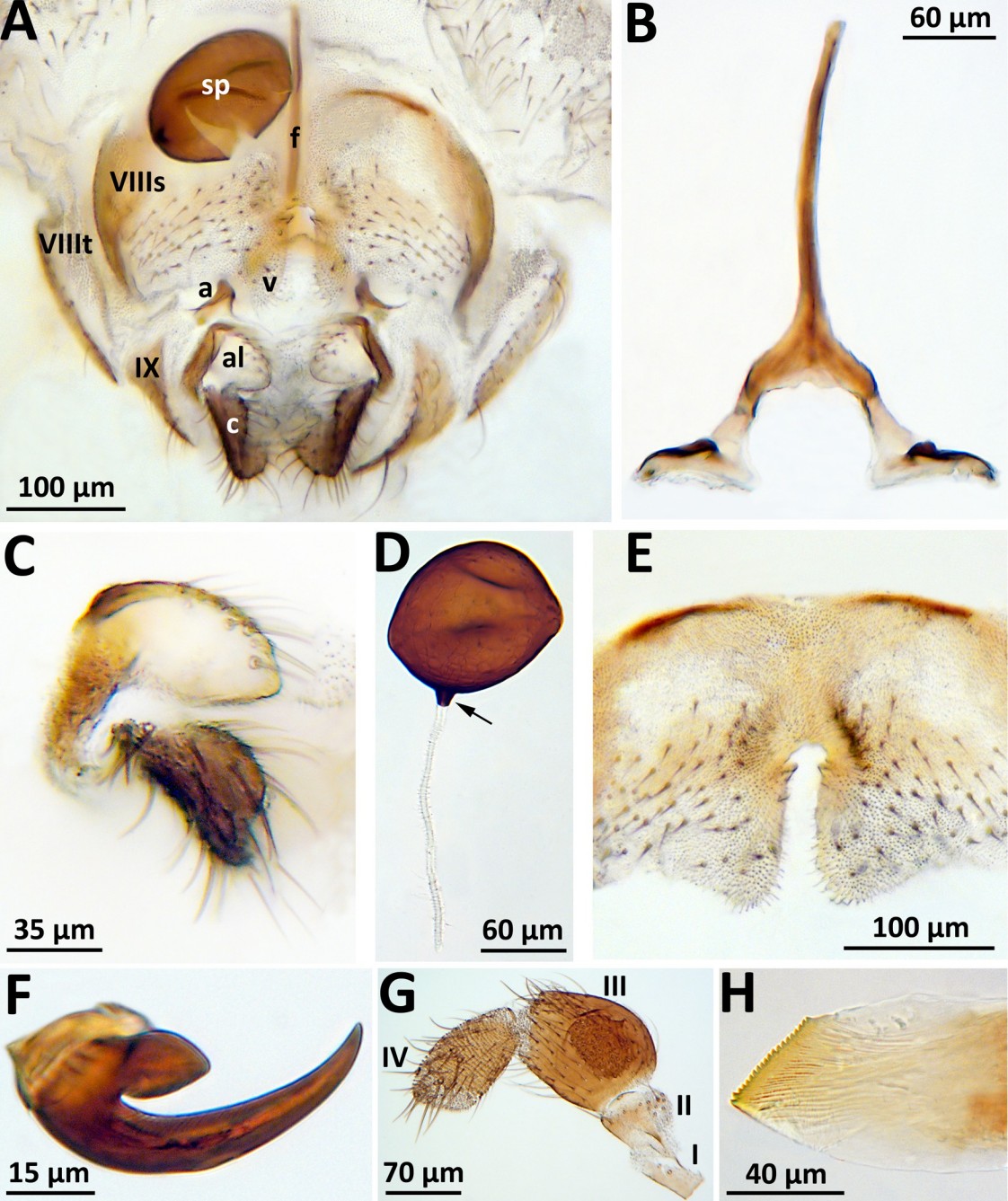

**Fig 1. Female of *Simulium mellah* (Spain, Turrillas, 24 June 2018).** (a) Terminalia, ventral view; a, anteriorly directed apodeme of lateral arm of genital fork; al, anal lobe; c, cercus; f, stem of genital fork; sp, spermatheca; v, hypogynial valve; IX, tergite of abdominal segment IX; VIIIs and VIIIt, sternite and tergite, respectively, of abdominal segment VIII; (b) Genital fork, ventral view; (c) Right anal lobe and cercus, ventral view; (d) Spermatheca; arrow indicates pigmented extension onto base of spermathecal duct; (e) Abdominal segment VIII with hypogynial valves, ventral view; (f) Claw of hind leg; (g) Maxillary palp; segments I–IV only (V not shown); (h) Apex of mandible.

serrations. The serrations in the *S. aureum* group are only on the inner margin of the mandible, as opposed to both margins, which is the condition in many other groups of black flies.

The two pharate males had a ventral plate (Fig 2E) identical to that originally described for *S. mellah* by Giudicelli et al. [47]: a narrow body with long basal arms (Fig 2E, 2G, 2H and 2I) versus a broader, shorter body with short basal arms in *S. rubzovianum /velutinum*. The

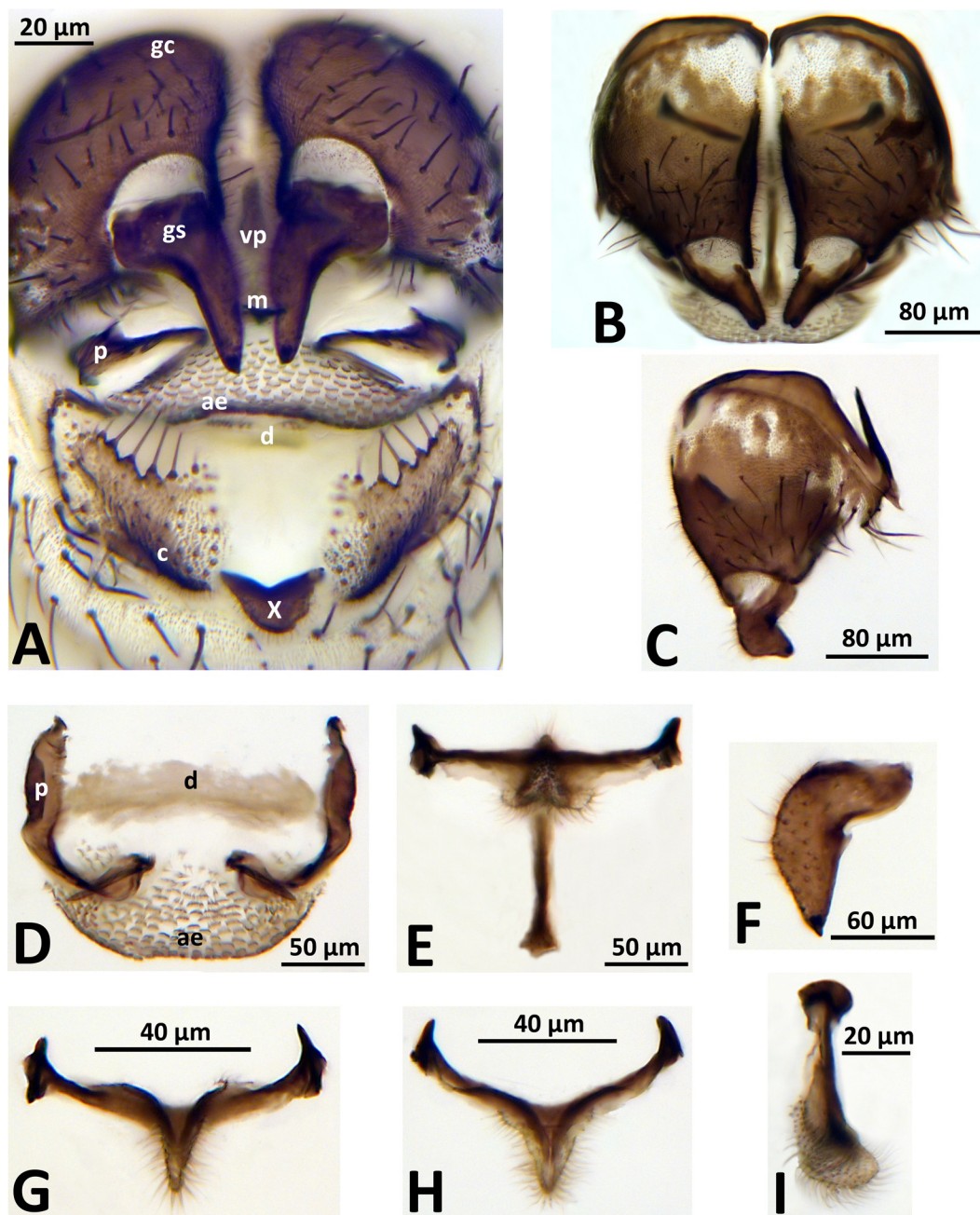

**Fig 2. Male of *Simulium mellah* (Spain, El Cautivo, 21 May 2019).** (a) Terminalia, terminal view; ae, aedeagal membrane; c, cercus; d, dorsal plate; gc, gonocoxa; gs, gonostylus; m, median sclerite; p, paramere; vp, ventral plate; X, tergite of abdominal segment X; (b) Genitalia, ventral view; (c) Right gonocoxa and gonostylus, inner lateral view; (d) Aedeagal membrane (ae), dorsal plate (d), and paramere (p) with parameral spine; terminal view; (e) Ventral plate and median sclerite, terminal view; (f) Gonostylus, dorsal view. G. Ventral plate, dorsal view; (h) Ventral plate, ventral view; (i) Ventral plate, lateral view.

gonostylus of *S. mellah* in inner lateral view is slightly more sharply angled at its midpoint (Fig 2C and 2F), compared with that of *S. rubzovianum*/*velutinum*, as shown by Crosskey & Crosskey [37], in agreement with the figure presented by Belqat & Dakki [48]. Orientation greatly influences the appearance of the gonostylus and can cause misidentification if comparisons of species are not based on identical orientations (Fig 2A–2C and 2F). Other male genitalic structures (Fig 2A, 2C and 2D) were typical of those of the *S. aureum* group, including the aedeagal membrane, dorsal and median sclerites, gonocoxites, and parameres.

For larvae of *S. mellah*, the diagnostic character given by Giudicelli et al. [47]—a dark head capsule—was insufficient for distinguishing *S. rubzovianum* and *S. mellah*. Chromosomal analyses revealed that all analyzed larvae (n = 32) were *S. mellah*, regardless of whether the head capsule was dark or light (Fig 3A and 3B). Larvae with pale head capsules resemble those of species such as *S. angustipes*, *S. rubzovianum*, and *S. velutinum*, whereas larvae with dark head capsules resemble those of *S. petricolum*. The shape of the postgenal cleft typically widened slightly before the apex in our specimens (Fig 3C and 3D) and in the illustration of material from the type locality by Giudicelli et al. [47]. In other members of the *S. aureum* group, the cleft is typically parallel-sided. This character, however, was subtle and therefore of limited taxonomic value. The hypostoma, traditionally a source of diagnostic characters for many black flies, did not provide a means of identifying *S. mellah* (Fig 3E and 3F).

The cocoon of *S. mellah* was shoe-shaped with an anteroventral collar (Fig 4E, whereas the cocoons of all other members of the *S. aureum* group are slipper-shaped without a collar [50]. The cocoons of *S. mellah* also were rather flimsy, with a looser weave and weaker anterior margin (Fig 4D and 4E) than those in other members of the *S. aureum* group. The shape of the four gill filaments (Fig 4A and 4B) and the arrangement of cephalic surface sculpture (Fig 4C) of the pupa of *S. mellah* were typical for the *S. aureum* group and did not provide characters sufficient for species identification.

Thus, the pupae, females, and males of *S. mellah* can be morphologically distinguished from those of all other members of the *S. aureum* group in Spain by one or more diagnostic characters (Table 1).

## 3.2. Chromosomes of the *Simulium aureum* group

The chromosomes of all 32 analyzed larvae agreed with the fixed banding pattern of larvae of *S. mellah* previously reported from Algeria and Morocco, including specimens from within 75 km of the type locality [6]. Thus, in contrast to other species of the *S. aureum* group (Table 1), *S. mellah* was fixed for inversions *IS-10*, *IS-11*, *IL-50*, and *IIL-20* (Table 2). One female larva carried a single, novel polymorphism: IIL-52 (Fig 5). We did not have larvae from the area where adults of *S. rubzovianum* and/or *S. velutinum* were trapped to chromosomally distinguish these two species or determine if closely related species, e.g., *S. aureum* cytospecies 'K' [6], were present.

## 3.3. Phylogeny and barcoding gap of the community of simuliids

**3.3.1. Phylogeny.** Phylogenetic analysis based on the COI region showed three main groups of black flies, one including *S. intermedium*; one including *S. pseudequinum*; and another including *S. angustipes*, *S. petricolum*, *S. rubzovianum*, and *S. mellah* (Fig 6A). In the first two groups, *S. intermedium* and *S. pseudequinum* were differentiated in two clades that matched the species identifications and clustered with sequences of the taxa retrieved from GenBank. In the third group, all individuals of *S. angustipes* clustered in a highly supported clade (98.9%) separated from *S. petricolum*, *S. rubzovianum*, and *S. mellah*. Individuals of *S. petricolum* all clustered in a subclade. However, individuals of *S. mellah* and *S. rubzovianum* were distributed throughout the clade (Fig 6B).

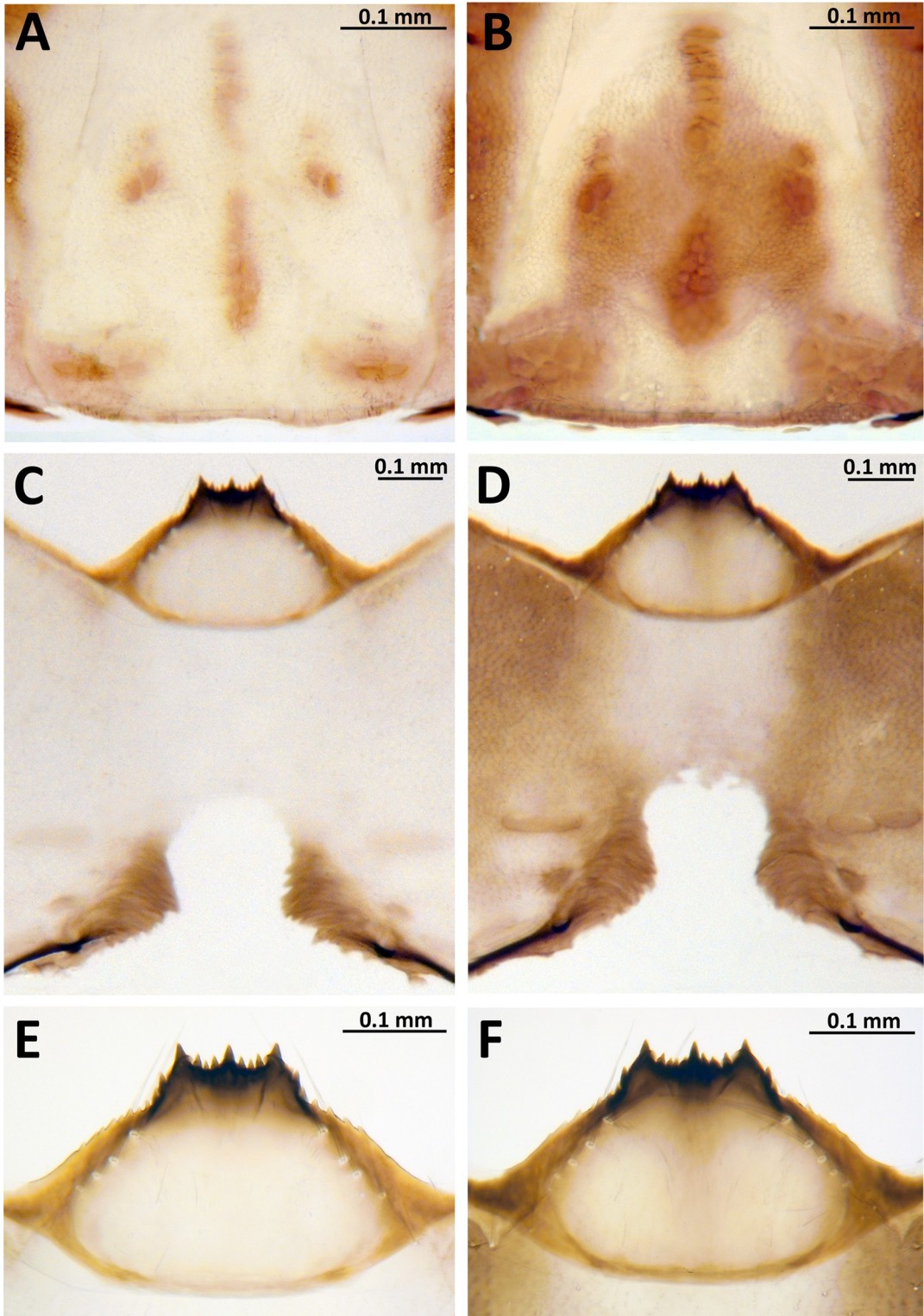

**Fig 3. Larva of *Simulium mellah* (Spain, El Cautivo, 7 April 2020).** (a, b) Head capsule, dorsal view; (c, d) Head capsule, ventral view; (e, f) Hypostoma; (a, c, e) Light form; (b, d, f) Dark form.

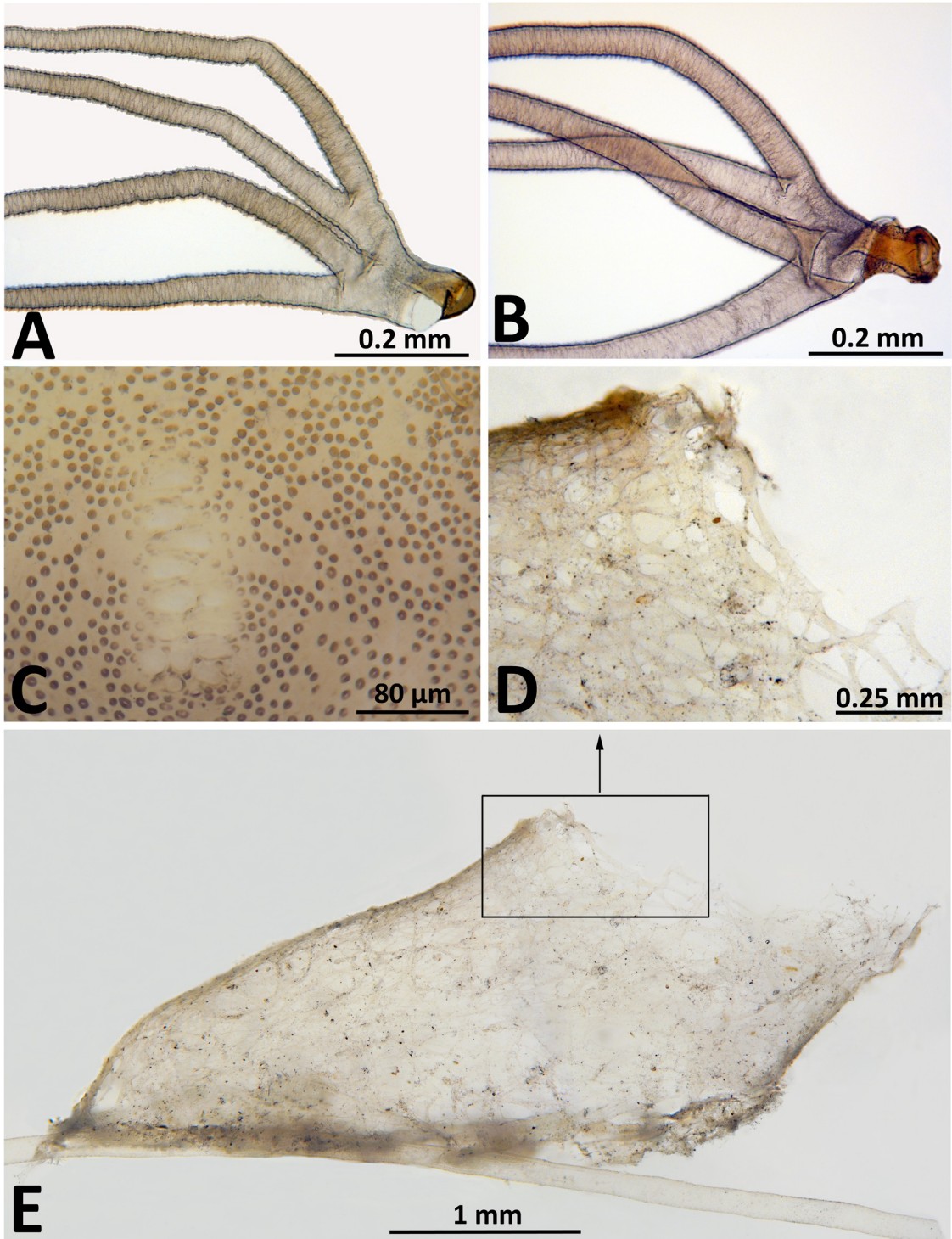

**Fig 4. Pupa and cocoon of *Simulium mellah* (Spain, El Cautivo, 21 May 2019).** (a, b) Gill base, showing variation, lateral view; (c) Middle of cephalic plate (male); (d) Anterior margin of cocoon, lateral view; (e) Cocoon, lateral view; box indicates area enlarged in d.

**Table 1. Diagnostic characters for *Simulium mellah* versus other species of the *S. aureum* group in Spain.**

| Stage (sex) | *S. mellah* | Other Spanish spp. of *S. aureum* group | Comments |
|---|---|---|---|
| Larva | Chromosomes with only the following fixed inversions: *IS-10*, *IS-11*, *IL-50*, and *IIL-20*. | Chromosomes either without *IL-50* and *IIL-20*, or if present, then with fixed inversions *IS-40* and *IIS-13*. | Chromosomal evaluation requires larvae fixed in Carnoy's solution. |
| Pupa | Cocoon shoe-shaped, with anteroventral collar. | Cocoon slipper-shaped, without anteroventral collar. | Care is required during removal of the pupa and cocoon from the substrate to ensure that the collar is not torn. |
| Male | Ventral plate in ventral view with basal arms longer than body, which is a short narrow triangle. | Ventral plate in ventral view with basal arms as long as or shorter than body, which is a short equilateral triangle or an elongate slender triangle. | Orientation in strict ventral view is critical. |
| Female | Spermathecal duct pigmented basally. Inner margins of hypogynial valves rather parallel sided with intervening space narrow. | Spermathecal duct unpigmented basally, or if pigmented, then inner margins of hypogynial valves divergent with intervening space 'U' or 'V' shaped. | Viewing the spermatheca, but not the hypogynial valves, requires clearing of the abdomen. |

Phylogenetic analysis based on the ITS2 region clustered individuals of the same species (Fig 7), matching the morphological identifications. *Simulium petricolum*, *S. rubzovianum*, and *S. mellah* clustered in a highly supported clade (88.2%) separated from *S. intermedium* and *S. pseudequinum*. The clade with *S. petricolum*, *S. rubzovianum*, and *S. mellah* included two highly supported subclades (98.9%), one of which included all three species in which *S. petricolum* and *S. mellah* were in a highly supported clade (88.9%), and one of which included only individuals of *S. rubzovianum* (Fig 7). Owing to ITS2 congruence, we were able to identify four of the five unknown COI sequences obtained from larvae (OQ469690, OQ469691, OQ469692 and OQ469694) (Fig 6B and S1 Table) as *S. rubzovianum* (Fig 7 and S2 Table). One COI larval sequence remains unidentified (OQ469693) because ITS2 amplification failed.

**3.3.2. Barcoding gap.** Based on COI sequences, we found 127 haplotypes, with haplotype diversity of 0.998 and 184 polymorphic sites. Nucleotide diversity was 0.047. Intraspecific distance was 0.0–8.08% and interspecific distance was 0.0–18.30%. We found contrary threshold values, depending on the method: a threshold of 6.11% based on "localMinima" and a threshold of 0.6% based on "threshOpt" segregating intra- from interspecific distances (S1 and S2 Figs). As "localMinima" could fail to estimate threshold values if intraspecific distances are greater than interspecific distances, the second value (0.6%) was set as the threshold. Based on that threshold, 91 of 140 individuals were identified correctly (0.65), 39 were unidentified (0.28), four were ambiguously identified (0.02), and six were incorrectly identified (0.04) by COI sequences (Table 3). *Simulium mellah* was the species most often incorrectly identified (0.67, four out of six individuals). Barcoding gap analysis by *TaxonDNA* set a threshold of 2.36% and led to better identifications, namely 127 correct identifications (0.91), seven

**Table 2. Frequency of chromosomal rearrangements in larvae of *Simulium mellah* from Spain, Almería province, Tabernas, El Cautivo.**

| Date | 13 Nov 2019 | 7 Apr 2020 |
|---|---|---|
| Females: males | 5:3 | 12:12 |
| *IS-10*\* | 1.00 | 1.00 |
| *IS-11* | 1.00 | 1.00 |
| *IL-50* | 1.00 | 1.00 |
| *IIL-20* | 1.00 | 1.00 |
| IIL-52 | 0.12 | 0.00 |

\* Italicized inversions are fixed; non-italicized inversion is polymorphic.

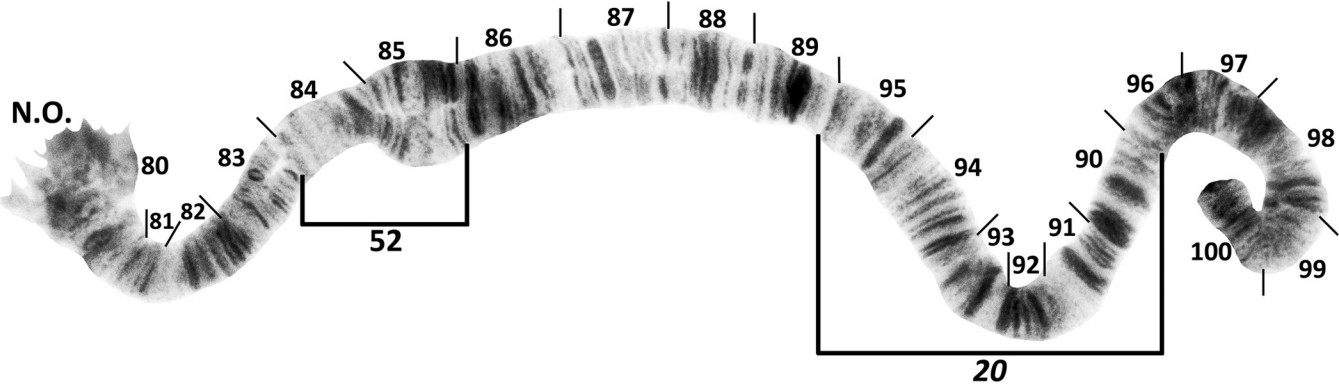

**Fig 5. IIL chromosomal sequence of *Simulium mellah* (female larva, Spain, El Cautivo, 13 November 2019), showing fixed inversion *IIL-20* and limits of polymorphic inversion IIL-52.**

ambiguous (0.05), and six incorrect identifications (0.04). In addition, we also found higher maximum intraspecific distances than minimum interspecific distances (mean ± SD = 3.00 ± 1.67 vs 2.88 ± 4.61; Fig 8).

Based on ITS2 sequences, we found 19 haplotypes, with haplotype diversity of 0.713 and 148 polymorphic sites. Nucleotide diversity was 0.026. Intraspecific distance was 0.0–1.47% and interspecific distance was 1.16–25.32%. We found several threshold values based on "localMinima" (S3 Fig), selecting 0.92% due to the concordance with the "threshOpt" results (S4 Fig). Based on that threshold, all 124 individuals were identified correctly by ITS2

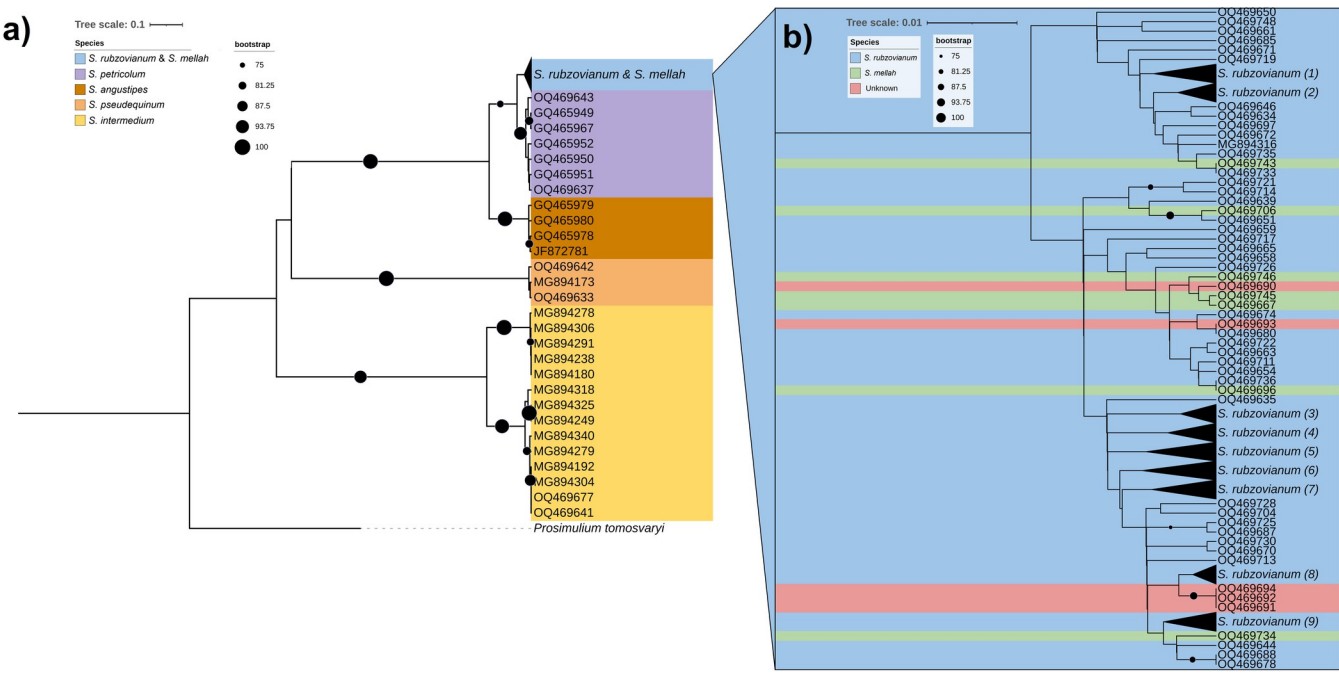

**Fig 6.** (a) Phylogenetic tree based on COI sequences, with maximum likelihood topology, of black flies from southeastern Spain, together with sequences from GenBank. The bootstrap values are only shown on nodes with bootstrap values higher than 75%, indicated by black circles in the figure, and range from 75% to 100% depending on their size. (b) Zoom-in of the group formed by *S. rubzovianum* and *S. mellah*. Unknown individuals correspond to larvae of subgenus *Eusimulium* which were not identified to species level prior to DNA analysis. Clusters with a value in brackets at the end of the species mean that these sequences have been collapsed and the information on the component species is in S1 Table. All sequences used for the phylogeny are summarized in S1 Table.

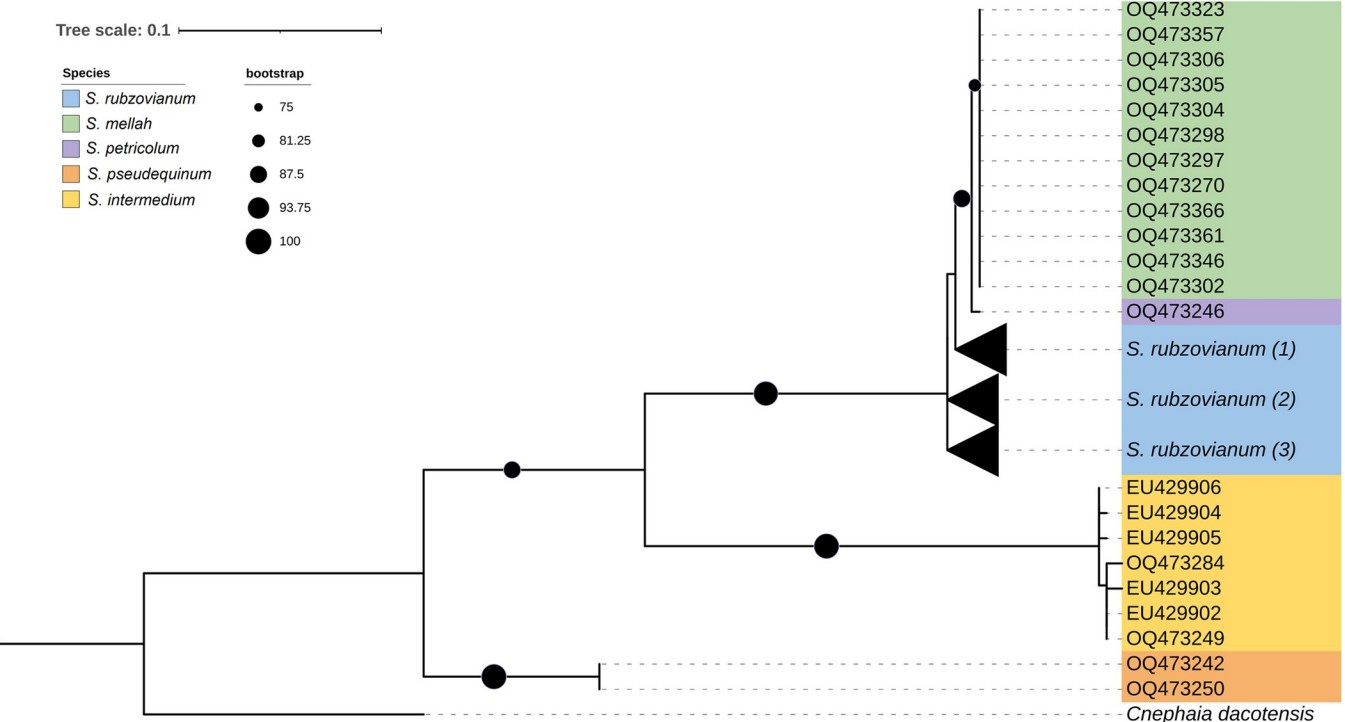

**Fig 7. Phylogenetic tree based on ITS2 sequences, with maximum likelihood topology, of black flies from southeastern Spain, together with sequences from GenBank.** The bootstrap values are only shown on nodes with bootstrap values higher than 75%, indicated by black circles in the figure, and range from 75% to 100% depending on their size. The unknown larvae were assigned to the closest group due to the high resolution and congruence shown by ITS2. Clusters with a value in brackets at the end of the species mean that these sequences have been collapsed and the information on the component species is in S2 Table. All sequences used for the phylogeny are summarized in S2 Table.

sequences (Table 3). Barcoding gap analysis by *TaxonDNA* set a threshold of 0.61% and lead to less accuracy, namely 119 correct identifications (0.96) and five without identifications (0.04). Furthermore, we found lower maximum intraspecific distances than minimum interspecific distances (mean ± SD = 0.50 ± 0.28 vs 2.94 ± 5.46; Fig 8). Therefore, ITS2 confirmed the presence of larvae and pupae of *S. rubzovianum/velutinum* and *S. mellah*.

**Table 3. Mean and ranges of intra- and interspecific distances per species and DNA region and identification status of black flies from southeastern Spain, based on "best close match".** For COI sequences, given a threshold of 0.6% obtained by "threshOpt" and a threshold of 2.36% based on *TaxonDNA*. For ITS2 sequences, given a threshold of 0.92% obtained by "threshOpt" and a threshold of 0.96% based on *TaxonDNA*.

| Species | Gene (n) | Mean intraspecific distance (range) | Mean interspecific distance (range) | Identification (*threshOpt* threshOpt threshold/ *TaxonDNA* threshold) | | | |
|---|---|---|---|---|---|---|---|
| | | | | correct (%) | incorrect (%) | ambiguous (%) | no id (%) |
| *S. rubzovianum* | COI (106) | 1.50% (0–3.27) | 10.27% (0–18.30) | 67/91 | 2/2 | 3/7 | 28/0 |
| | ITS2 (109) | 0.16% (0–0.77) | 14.25% (1.16–24.54) | 100/100 | 0/0 | 0/0 | 0/0 |
| *S. mellah* | COI (6) | 1.86% (0.16–2.93) | 4.02% (0–18.30) | 33/33 | 67/67 | 0/0 | 0/0 |
| | ITS2 (6) | 0% (0–0) | 3.17% (1.16–25.32) | 100/100 | 0/0 | 0/0 | 0/0 |
| *S. pseudequinum* | COI (3) | 0.85% (0.48–1.12) | 16.26% (15.25–17.87) | 67/100 | 0/0 | 0/0 | 33/0 |
| | ITS2 (2) | 0% (0–0) | 22.41% (21.71–24.18) | 100/100 | 0/0 | 0/0 | 0/0 |
| *S. intermedium* | COI (14) | 4.22% (0–8.08) | 16.38% (14.26–18.30) | 93/100 | 0/0 | 0/0 | 7/0 |
| | ITS2 (7) | 0.69% (0–1.47) | 23.02% (21.91–25.32) | 100/29 | 0/0 | 0/0 | 0/71 |
| *S. angustipes* | COI (4) | 0.88% (0.48–1.28) | 1.60% (0.16–2.43) | 50/100 | 0/0 | 0/0 | 50/0 |
| *S. petricolum* | COI (7) | 1.60% (0.16–2.43) | 0.88% (0.48–1.28) | 29/100 | 0/0 | 0/0 | 71/0 |

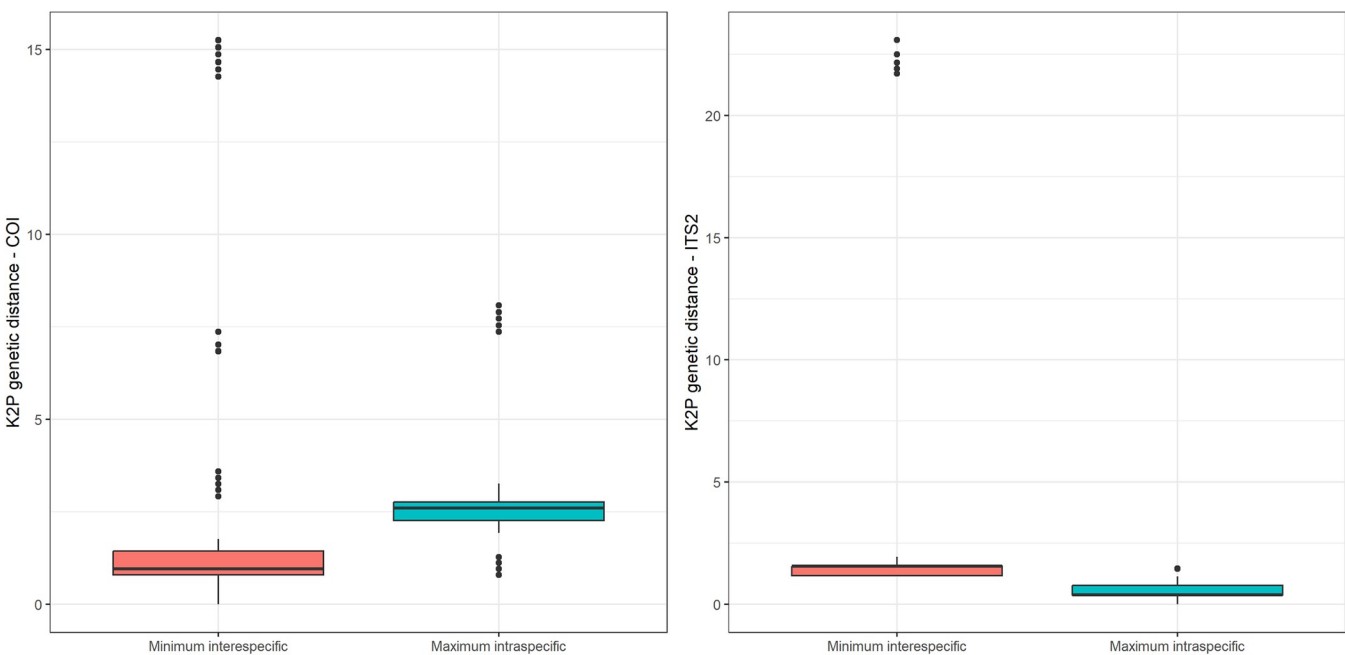

**Fig 8. Boxplot of minimum interspecific and maximum intraspecific genetic distances per sequence from all COI and ITS2 sequences of black flies.** The median is indicated by the line inside the box. The lower and upper hinges correspond to the first and third quartiles (the 25th and 75th percentiles). The upper whisker extends from the hinge to the largest value no farther than 1.5 * IQR from the hinge (where IQR is the inter-quartile range, or distance between the first and third quartiles). The lower whisker extends from the hinge to the smallest value, at most 1.5 * IQR of the hinge. Data beyond the end of the whiskers are outlying points and are plotted individually.

## 4. Discussion

We present the first comprehensive study of the simuliid community in a semi-arid ecosystem of Iberia, encompassing a variety of trapping methods, identification techniques, life stages, and habitat types. Our study confirms the presence of five species (*S. intermedium*, *S. petricolum*, *S. pseudequinum*, *S. mellah*, and *S. rubzovianum*) in Almería province and provides the first report of *S. mellah* for Europe.

Four of these species have previously been reported from Almería province [35, 37, 68]. *Simulium intermedium*, *S. pseudequinum*, and *S. rubzovianum* are common, with circum-Mediterranean distributions [5, 69] *Simulium intermedium* is abundant in Spain and Portugal, often in streams degraded by sewage effluent [37]. *Simulium pseudequinum* is abundant in Andalusian rivers and large streams (sometimes in aqueducts) below 800 m asl. *Simulium petricolum* is considered rare, compared with the common and widespread *S. rubzovianum* (as *S. velutinum*) in Andalusia and Spain [36, 37].

### 4.1. *Simulium mellah*, a new species record for Europe

An integrated taxonomic approach applied to larvae, pupae, males, and females allowed us to identify *S. mellah* and evaluate its diagnostic characters across a broad geographic range. Our study extends the distribution of this species from North Africa to Europe by 760 km between the nearest sites on the two continents. *Simulium mellah* becomes the 43rd species of black fly shared between the African and European continents [5].

The taxonomic characters that we found for *S. mellah* indicate more intraspecific variation than in the original description, which was based on a single location. Local adaptation, phenotypic plasticity, and environmental conditions can influence the characteristics of species. For

instance, we found that breeding sites of *S. mellah* in our study are at lower elevations (240–470 m asl) than in North Africa (480–1400 m asl) [30, 47]. Moreover, the density of cocoon weave can be diagnostic in our material of *S. mellah*, but in some species it varies with current velocity [70]. Similarly, larval body colour and intensity of pigmentation of the head capsule and head spots can be diagnostic in some species of black flies, but these characters can be influenced by diet, sunlight, sex, and substrate to which the larvae are attached [71, 72]. In the *S. aureum* group, larvae of *S. rubzovianum* and *S. velutinum* have light head capsules, whereas our larvae of *S. mellah* have dark or light heads; the degree of pigmentation is not related to larval sex. Although we present diagnostic morphological characters to distinguish the pupae, males, and females of *S. mellah* from all other members of the *S. aureum* group, the polytene chromosomes or ITS2 gene are needed to identify the larvae.

## 4.2. Taxonomic complexity of black flies and tools to address the complexity

We found that the frequently used COI region is not adequate to identify all species in our study, whereas ITS2 correctly identified all of them. Phylogenetic analysis based on the COI region segregates some clades of black flies (*S. intermedium*, *S. pseudequinum*, and *S. angustipes*) but does not resolve relationships of *S. petricolum*, *S. rubzovianum*, and *S. mellah*. COI also fails to infer relationships of certain closely related nominal species in species groups such as the *S. variegatum* and *S. vernum* groups in Europe [9] and the *S. multistriatum* and *S. striatum* groups in Thailand [7, 8]. Barcoding gap analysis indicates that COI is worse than ITS2 for identifying the species in our study, but with important differences depending on the threshold calculation method. Furthermore, intraspecific genetic distances were greater than interspecific distances, making the COI gene inadequate to identify the species in our study area.

Phylogenetic analysis using the ITS2 region clusters *S. petricolum*, *S. rubzovianum*, and *S. mellah* together but separates them into respective sister clades. The ITS2 region is a fast-evolving intergenic spacer that allows inference of relationships between closely related species [73]. Barcoding gap analysis shows that the ITS2 region successfully identified all individuals, and intraspecific genetic distances were lower than interspecific distances, making this marker a good candidate for differentiating these species. Nevertheless, ITS2 is prone to insertions or deletions, complicating the alignment, which together with its faster-evolving rate, could affect the resolution of species. Thus, multiple genetic markers, such as the elongator complex protein 1 gene (ECP1) [74] or 16S [75], should also be used to infer simuliid phylogeny. Most of the sequences we explored belong to the same spatial region and, for some species, the number of sequences is low. Further studies exploring the utility of ITS2 as a molecular marker to identify black flies are needed, including more individuals per species from more locations to ensure resolution under higher intraspecific variance.

Due to the scarcity of ITS2 sequences of the Simuliidae in the databases, combining both techniques (COI and ITS2) is advisable. For instance, Chakarov et al. [32] identified the black flies captured in nest boxes in their study as *S. rubzovianum*, based on the COI gene. In light of our findings, it would be of value to use the ITS2 region to determine if any specimens of *S. mellah* were misidentified by COI as *S. rubzovianum*.

## 4.3. Biodiversity and ecology of black flies in a semi-arid environment

*Simulium rubzovianum* was the most frequently captured species in nest boxes and CDC traps, suggesting that it is the most common simuliid in our study area. The identification of *S. rubzovianum*, however, carries possible complications. The species long known as *S. velutinum*

and once considered widespread throughout Africa, Europe, and Western Asia is actually composed of multiple species, one of which is *S. rubzovianum* [6]. We have considered the specimens in our study to be *S. rubzovianum* because most larvae that have been chromosomally identified in Spain are this species. However, larvae of *S. velutinum s.s.* have been found in La Rioja province (42˚34'42"N 02˚51'13"W) (Ruiz-Arrondo & Adler, unpublished data). Further complicating the taxonomy of *S. rubzovianum* and *S. velutinum s.s.* is the possible presence of undescribed species [6]. The presence of three distinct, highly supported clades of *S. rubzovianum* in our ITS2 tree suggests that *S. velutinum s.s.*, or undescribed species, could be in our study area.

Of the five species identified in our area, only two have previously been found in roller nest boxes [32, 40]. We add a third species (*S. mellah*). Unlike *S. rubzovianum*, in which roller blood has been found in female flies [32], caution is warranted for *S. mellah* because none have been evaluated for roller blood. *Simulium petricolum* is one of the two species previously identified from roller nest boxes (1 female) in our study area [40]. However, we did not find this species in nest boxes, nor was it found by Chakarov et al. [32]. We suspect that the individual identified as *S. petricolum* by Václav et al. [40] is *S. mellah*.

The other two species identified in our study, *S. intermedium* and *S. pseudequinum*, are mammalophilic and typically feed on equids and ruminants that are not abundant in the study area; thus, they probably fed on other mammals such as goats and sheep. Further studies should evaluate the hosts of these mammalophilic species and the feeding habits of *S. mellah* and its possible role in transmitting avian blood parasites. Identifying which species of black flies feed on open-nesting avian species (e.g., some species of doves and pigeons), would require another type of trapping method, such as scoop netting [24].

*Simulium rubzovianum* (previously identified as *S. velutinum* in Spanish studies) is an abundant ecological generalist in the Iberian Peninsula [36, 69] where it has been reported to tolerate eutrophication and mineralization of water [69]. *Simulium mellah* ("mellah" meaning salt in Arabic), on the contrary, is a specialist restricted to small, saline streams. It was originally described from highly mineralized, salt-rich water in Morocco [47]. Subsequently discovered breeding sites in Algeria and Morocco also have high levels of salinity (e.g., 3.6%) [30]. Other species can breed in saline habitats, including *Simulium salinum* (Rubtsov) in Siberia [76], *Simulium vittatum* Zetterstedt on the maritime islands of Canada [72], and *Simulium lundstromi* (Enderlein) and *Simlium ruficorne* Macquart in Algeria [30]. Other semi-arid regions in southern Spain with saline water, such as Rambla Salada in Murcia [77], should be explored for *S. mellah* and other simuliids. For instance, *Simulium ruficorne*, a characteristic species of desert environments in North Africa and the Iberian Peninsula [31], might be expected in our study area or similar ramblas.

The breeding sites in our study represent an extreme habitat for the Simuliidae, based on characteristics such as low current, seasonal drought, high temperature, and high salinity. Extreme habitats can minimize competition, predation, and parasitism, as with members of the genus *Gymnopais* Stone [72]. Further ecological studies are required to investigate the effect of the characteristics of such a harsh habitat on the different life stages of the identified species.

## 5. Conclusion

The requirement of simuliids for flowing water led us to predict low simuliid richness in a semi-arid environment. Yet, we recorded at least five species in a relatively small area, which implies that other semi-arid regions in the Iberian Peninsula (and elsewhere) might harbor unexpected diversity. The finding of a new continental record (*S. mellah*) highlights the value

of studying macroinvertebrates in poorly studied habitats such as temporal watercourses in arid areas.

Members of certain species groups of black flies are especially difficult to identify morphologically, such as those of the *S. aureum* group. ITS2 is a promising marker to use together with COI for simuliid identification but a search of different markers also should be undertaken. We stress the importance of integrating different techniques to evaluate and identify the members of the black fly community. For the Simuliidae, molecular techniques should be integrated with morphology, cytology, and ecology.

## Supporting information

**S1 Table. COI sequences of *Simulium* species used in phylogenetic analysis, their accession numbers, labels in each phylogeny (Fig 6), species identifications, and origins and sources.** Clusters with a value in brackets at the end of the species (Fig 6) mean that these sequences have been collapsed and the information on the component species is in S1 Table.
(XLSX)

**S2 Table. ITS2 sequences of *Simulium* species used in phylogenetic analysis, their accession numbers, labels in the phylogeny (Fig 7), species identifications, and origins and sources.** Clusters with a value in brackets at the end of the species (Fig 7) mean that these sequences have been collapsed and the information on the component species is in S2 Table.
(XLSX)

**S1 Fig. Density plot based on COI sequences to determine the threshold genetic distance for species identification.** The transition between intra- and interspecific distances is the dip in the density graph, here approximately at 6% or 11%.
(TIF)

**S2 Fig. Barplot based on COI sequences showing the percentage of cumulative error (false positive (light grey) plus false negative (dark grey)) of identification of black fly species with a pre-set threshold change (threshold values from 0.1% to 2% in 0.1% increments).** Optimum threshold is between 0.5% and 0.7%.
(TIF)

**S3 Fig. Density plot based on ITS2 sequences to determine the threshold genetic distance for species identification of black flies.** The transition between intra- and interspecific distances is the dip in the density graph. Here several thresholds appear, at low K2P distances, namely 0.12%, 0.52%, 0.92%, 1.27%, and 1.72%.
(TIF)

**S4 Fig. Barplot based on ITS2 sequences showing the percentage of cumulative error (false positive (light grey) plus false negative (dark grey)) of identification of black fly species, with a pre-set threshold change (threshold values from 0.1% to 2% in 0.1% increments).** Optimum threshold is between 0.8% and 1.1%.
(TIF)

**S1 File. Abstract in Spanish.**
(DOCX)

## Acknowledgments

We thank Stanislav Kolencik and Teresa Martínez for their help with the fieldwork. We are grateful to the authority of Junta de Andalucia for granting us permits.

## Author Contributions

**Conceptualization:** Ignacio Ruiz-Arrondo, Peter H. Adler, Francisco Valera.

**Formal analysis:** Ignacio Ruiz-Arrondo, Jesús Veiga, Peter H. Adler.

**Funding acquisition:** José A. Oteo, Francisco Valera.

**Methodology:** Ignacio Ruiz-Arrondo, Jesús Veiga, Peter H. Adler.

**Project administration:** Francisco Valera.

**Supervision:** José A. Oteo, Francisco Valera.

**Writing – original draft:** Ignacio Ruiz-Arrondo, Jesús Veiga, Peter H. Adler, Francisco Valera.

**Writing – review & editing:** Ignacio Ruiz-Arrondo, Jesús Veiga, Peter H. Adler, Francisco Collantes, José A. Oteo, Francisco Valera.

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
