## [Decision Letter · Decision Letter 0]

6 Aug 2023

PONE-D-23-21797Integrated taxonomy of black flies (Diptera: Simuliidae) reveals unexpected diversity in the most arid ecosystem of EuropePLOS ONE

Dear Dr.  Ruiz-Arrondo,

Thank you for submitting your manuscript to PLOS ONE. After careful consideration, we feel that it has merit but does not fully meet PLOS ONE’s publication criteria as it currently stands. Therefore, we invite you to submit a revised version of the manuscript that addresses the points raised during the review process.

I list of the very minor points that need to be attended to is provided in the reviewers´ comments. Please also try to reduce the length of the manuscript as it is presently a little wordy.

We look forward to receiving your revised manuscript.

Kind regards,

James Lee Crainey, Ph.D.

Academic Editor

PLOS ONE

Journal Requirements:

4. Please expand the acronym “MCIN/AEI” (as indicated in your financial disclosure) so that it states the name of your funders in full.

6. We are unable to open your Supporting Information file Supporting information.7z. Please kindly revise as necessary and re-upload.

Reviewers' comments:

Reviewer's Responses to Questions

**Comments to the Author**

1. Is the manuscript technically sound, and do the data support the conclusions?

Reviewer #1: Yes

Reviewer #2: Yes

2. Has the statistical analysis been performed appropriately and rigorously? 

Reviewer #1: Yes

Reviewer #2: Yes

3. Have the authors made all data underlying the findings in their manuscript fully available?

Reviewer #1: Yes

Reviewer #2: Yes

4. Is the manuscript presented in an intelligible fashion and written in standard English?

Reviewer #1: Yes

Reviewer #2: Yes

5. Review Comments to the Author

Reviewer #1: This study using morphology, cytology and molecularly (COI and ITS2 sequences) to examine diversity of black flies in Tabernas Desert, Spain. The information provided in the present study is very interesting. I have some comments for author to consider.

- Line 376, barcoding gap, because the barcode gap analyses based on localMinima and threshOpt methods revealed two very different values (>10x differentiations). It might be useful to use additional method such as those available in TaxonDNA software (Meier et al., 2006) which author used for Best Close Match (BCM) species identification in the present study. Because the successful identification based on the BCM method rely on the threshold value, thus, using inappropriate threshold value can hugely affect the species identification. Also, it will be useful to use additional method for species identification such as Best Match method in TaxonDNA.

- Line 386 – 387, It will be very useful to provide range of intraspecific and interspecific genetic distances for all species included in this study.

- Table 3, In my opinion, the table is easier to read if the same species is report within a single row for each gene. Therefore, I suggest that under the “Identification” column, it can be divided into four subheadings, i.e. correct (%), incorrect (%), ambiguous (%) and no id (%).

- Line 391, I am surprise that the number of haplotype is very low for the ITS2 sequences but the haplotype diversity value is fairly high. According to line 394, total specimens for the ITS2 sequences are 124 individuals but only 19 haplotypes were identified. However, the haplotype diversity reported is 0.713. Please check this again. Also, it will be useful to report range of intraspecific and interspecific genetic distance for the ITS2 sequences.

Reviewer #2: The authors used morphological, chromosomal, and molecular methods to investigate the systematics of black flies in a semi-arid area of the Iberian Peninsula. The rationale is that the Iberian Peninsula is atypical of where black flies breed, therefore, has not been adequately studied. Adult black flies were collected in different ecological habitats using CDC traps, sticky traps in man-made bird nests and pre-imago stages at breeding sites. The morphological and cytological methods employed standard methods and the molecular analysis targeted the mitochondrial cytochrome C oxidase subunit I and the internal transcribed spacer 2 (ITS2) that involved barcoding gap and phylogenetic analyses. Five species of black flies Simulium intermedium, S. petricolum, S. pseudequinum, S. rubzovianum and S. mellah were identified, the latter being the first to be recorded in Europe. The ITS2 was found to be the best for species identification and the establishment of the black flies' phylogenetic relationships.

The manuscript is technically sound and the data support the conclusions. The statistical analysis is OK except for the cytotaxonomic analysis where I was expecting the use of Hardy-Weinberg equation to estimate allele frequencies and therefore, the veracity of the S. mellah as a species population. The authors have fully complied with the data availability.

Comments.

I. Keywords. I suggest that 'Iberian Peninsula' 'black flies' and 'diversity' are inserted, and 'aridity' be removed. Both both and ITS2 be written in full.

2. In demonstrating diversity, it would have helped if the spatial distribution of the species in the area was shown.

6. PLOS authors have the option to publish the peer review history of their article (what does this mean?). If published, this will include your full peer review and any attached files.

Reviewer #1: No

Reviewer #2: No

---

## [Author Response · Author response to Decision Letter 0]

20 Sep 2023

Review Comments to the Author

Reviewer #1

This study using morphology, cytology and molecularly (COI and ITS2 sequences) to examine diversity of black flies in Tabernas Desert, Spain. The information provided in the present study is very interesting. I have some comments for author to consider.

Authors: Thank you for your positive feedback.

- Line 376, barcoding gap, because the barcode gap analyses based on localMinima and threshOpt methods revealed two very different values (>10x differentiations). It might be useful to use additional method such as those available in TaxonDNA software (Meier et al., 2006) which author used for Best Close Match (BCM) species identification in the present study. Because the successful identification based on the BCM method rely on the threshold value, thus, using inappropriate threshold value can hugely affect the species identification. Also, it will be useful to use additional method for species identification such as Best Match method in TaxonDNA.

Authors: Thanks for your valuable comment, we agree with the importance of the threshold value and we add an additional method of calculation based on TaxonDNA (please, see lines 279-282 in the version with Tracked changes). We found better identification results for COI but not for ITS2, which have been added to the new version of the manuscript (please, see lines 402-404, 469-471 and Table 3 in the version with Tracked changes). Regarding the use of Best Match, we chose not to include it in the manuscript because it assigns the species name from its barcode that best matches, regardless of how similar the query and barcode sequences are. Moreover, the results were quite similar to those obtained by Best Close Match, so we finally decided not to include it in order not to overload this section. However, if the reviewer or editor finally considers including it, we have no objection to include it.

- Line 386 – 387, It will be very useful to provide range of intraspecific and interspecific genetic distances for all species included in this study.

Authors: We agree with the reviewer. Accordingly, we provide the suggested information (please, see lines 393-394, 414-415 and Table 3 in the version with Tracked changes). We also noticed that the result of intra- and interspecific distances was not accurately described, as we want to refer to maximum intraspecific and minimum interspecific distances, which are key for barcoding gap analysis. We made some changes in the manuscript to correct it (please, see lines 404-405, 420 and Figure 8 and its caption in the version with Tracked changes).

- Table 3, In my opinion, the table is easier to read if the same species is report within a single row for each gene. Therefore, I suggest that under the “Identification” column, it can be divided into four subheadings, i.e. correct (%), incorrect (%), ambiguous (%) and no id (%).

Authors: We modified Table 3 as requested in order to see each species in a single row for ease of reading. We also added intra and interspecific distances to the table, as suggested in the previous comment (please, see Table 3).

- Line 391, I am surprise that the number of haplotype is very low for the ITS2 sequences but the haplotype diversity value is fairly high. According to line 394, total specimens for the ITS2 sequences are 124 individuals but only 19 haplotypes were identified. However, the haplotype diversity reported is 0.713. Please check this again. Also, it will be useful to report range of intraspecific and interspecific genetic distance for the ITS2 sequences.

Authors: We double-checked the result, and it is correct. Haplotype diversity, calculated using Nei and Tajima's (1981) method, is defined as one minus the sum of the squared relative frequencies of each haplotype, multiplied by the sample size divided by the sample size minus 1. 

In this formula, ‘pi’ is the (relative) haplotype frequency of each haplotype in the sample and n is the sample size (Fan et al. 2021).

Therefore, for the ITS2, we have 19 haplotypes with the following frequencies: 1, 1, 61, 1, 7, 2, 2, 1, 1, 24, 10, 5, 1, 1, 1, 1, 2, 1, 1. When we calculate one minus the sum of their squared relative frequencies (0,292), we obtain 0,7077, which multiplied by 124 divided by 123, we get 0,713.

We did not specify in the previous version the method followed for such calculations, which it is now detailed (please see lines 287-288 in the version with Tracked changes).

The results regarding haplotype diversity are intriguing and deserve deep exploration, but additional analysis and considerations are necessary to make accurate predictions.

References:

-Nei, M., & Tajima, F. (1981). DNA polymorphism detectable by restriction endonucleases. Genetics, 97(1), 145-163.

-Fan, P., Fjeldså, J., Liu, X., Dong, Y., Chang, Y., Qu, Y., ... & Lei, F. (2021). An approach for estimating haplotype diversity from sequences with unequal lengths. Methods in Ecology and Evolution, 12(9), 1658-1667.

Reviewer #2

The authors used morphological, chromosomal, and molecular methods to investigate the systematics of black flies in a semi-arid area of the Iberian Peninsula. The rationale is that the Iberian Peninsula is atypical of where black flies breed, therefore, has not been adequately studied. Adult black flies were collected in different ecological habitats using CDC traps, sticky traps in man-made bird nests and pre-imago stages at breeding sites. The morphological and cytological methods employed standard methods and the molecular analysis targeted the mitochondrial cytochrome C oxidase subunit I and the internal transcribed spacer 2 (ITS2) that involved barcoding gap and phylogenetic analyses. Five species of black flies Simulium intermedium, S. petricolum, S. pseudequinum, S. rubzovianum and S. mellah were identified, the latter being the first to be recorded in Europe. The ITS2 was found to be the best for species identification and the establishment of the black flies' phylogenetic relationships.

The manuscript is technically sound and the data support the conclusions. The statistical analysis is OK except for the cytotaxonomic analysis where I was expecting the use of Hardy-Weinberg equation to estimate allele frequencies and therefore, the veracity of the S. mellah as a species population. The authors have fully complied with the data availability.

Authors: Thank you very much for your positive feedback. 

We respectfully note that Hardy-Weinberg calculations cannot be applied to our data because the inversions (alleles) are fixed (i.e., 100%), except one polymorphic inversion (IIL-52), which was represented in only one homologue in one individual larva. Thus, the polymorphism data on which Hardy-Weinberg calculations would be performed are not possible, either because the inversions are not polymorphic or the single polymorphism represented in only one larva does not meet statistical requirements for calculations. We note that the chromosomal banding pattern, which is unique among all known species in the subgenus, is entirely consistent with all other known populations of S. mellah, including material from near the type locality; conclusions of conspecificity are, therefore, justified.

Comments.

I. Keywords. I suggest that 'Iberian Peninsula' 'black flies' and 'diversity' are inserted, and 'aridity' be removed. Both COI and ITS2 be written in full.

Authors: Thank you for the suggestion. We understand that "black flies" and "diversity" do not have to be included as keywords as they appear in the title of the manuscript and including them would be losing the opportunity to give importance to other key words. As suggested by the reviewer, "Iberian Peninsula" has been included, the COI and ITS2 acronyms have been developed and "aridity" has been removed.

2. In demonstrating diversity, it would have helped if the spatial distribution of the species in the area was shown.

Authors: The reviewer's suggestion is interesting, and we are willing to do it if the Editor thinks it is appropriate. Although the authors do not understand how the spatial distribution of species in our case (a study area of moderate size) can demonstrate diversity. Furthermore, we believe that the manuscript is already long and that doing what the reviewer requests would probably require several images (or one that is too complex). We understand that the other images included both in the main text and in the supplementary text are more important than this one.

---

## [Decision Letter · Decision Letter 1]

16 Oct 2023

Integrated taxonomy of black flies (Diptera: Simuliidae) reveals unexpected diversity in the most arid ecosystem of Europe

PONE-D-23-21797R1

Dear Dr. Ruiz-Arrondo,

We’re pleased to inform you that your manuscript has been judged scientifically suitable for publication and will be formally accepted for publication once it meets all outstanding technical requirements.

Kind regards,

James Lee Crainey, Ph.D.

Academic Editor

PLOS ONE

Reviewers' comments:

Reviewer's Responses to Questions

**Comments to the Author**

1. If the authors have adequately addressed your comments raised in a previous round of review and you feel that this manuscript is now acceptable for publication, you may indicate that here to bypass the “Comments to the Author” section, enter your conflict of interest statement in the “Confidential to Editor” section, and submit your "Accept" recommendation.

Reviewer #1: All comments have been addressed

2. Is the manuscript technically sound, and do the data support the conclusions?

Reviewer #1: Yes

3. Has the statistical analysis been performed appropriately and rigorously? 

Reviewer #1: Yes

4. Have the authors made all data underlying the findings in their manuscript fully available?

Reviewer #1: Yes

5. Is the manuscript presented in an intelligible fashion and written in standard English?

Reviewer #1: Yes

6. Review Comments to the Author

Reviewer #1: Thank you very much for a revised manuscript. All of my comments and suggestions were adequately responded.

7. PLOS authors have the option to publish the peer review history of their article (what does this mean?). If published, this will include your full peer review and any attached files.

Reviewer #1: No

---

## [Editor Report · Acceptance letter]

2 Nov 2023

PONE-D-23-21797R1 

Integrated taxonomy of black flies (Diptera: Simuliidae) reveals unexpected diversity in the most arid ecosystem of Europe 

Dear Dr. Ruiz-Arrondo:

I'm pleased to inform you that your manuscript has been deemed suitable for publication in PLOS ONE. Congratulations! Your manuscript is now with our production department. 

Kind regards, 

on behalf of

Dr. James Lee Crainey 

Academic Editor

PLOS ONE